# CollabMask: Explainable Neuron Collaboration Gradient Masks for LLM Fine-Tuning

## Abstract

The rapid advancement of large language models (LLMs) has increased the need for effective task-specific adaptation. Fine-tuning remains the primary approach, but it often suffers from redundant parameter updates. Existing methods mitigate these issues using gradient masks to constrain parameter updates, yet they largely ignore the interactions among neurons. We observe **neuron collaboration**, the phenomenon where groups of neurons are more likely to be co-activated to perform specific tasks. Leveraging this concept, we propose `CollabMask` (Collaborative Neuron Mask Fine-tuning), which constructs a co-activation hypergraph to capture neuron collaboration, clusters neurons into functional groups, and generates dynamic, collaboration- and function-aware gradient masks. By preserving collaborative patterns and prioritizing functionally important neurons, `CollabMask` improves task adaptation while retaining pretrained knowledge. Experiments on math, coding, and medical benchmarks show up to a 2.4% improvement over representative baselines, which demonstrates `CollabMask`'s ability to filter gradient noise and highlights the interpretability value of neuron collaboration groups.

## 1 Introduction

Large language models (LLMs) have advanced rapidly in recent years and are increasingly adapted to diverse application domains (Bubeck et al., 2023; Achiam et al., 2023; Thirunavukarasu et al., 2023; Wen et al., 2024). Fine-tuning pretrained models is commonly the dominant strategy for task specialization and thus remains a central focus of current research (Wang et al., 2024; Hu et al., 2021).

However, the increasing scale of pretrained models, coupled with the limited size of downstream datasets, poses significant challenges for fine-tuning (Zhang et al., 2024). The process of updating billions of parameters with gradients derived from batches of a limited training dataset can introduce noisy and redundant parameter updates, hindering the model's fine-tuning performance (Xu et al., 2021; Zhong et al., 2022; Li et al., 2025).

To mitigate these challenges, recent work has explored constraining parameter updates during fine-tuning. For instance, Child-Tuning (Xu et al., 2021) and HMT (Hui et al., 2024) apply random gradient masks to selectively update parameters in MLP layers, while GMT (Li et al., 2025) selects gradient masks based on the average gradient across several batches. LoRA (Hu et al., 2021) takes a different approach by restricting updates to low-dimensional parameter subspaces. The core principle of these approaches is to regularize the hypothesis space, thereby reducing overfitting and preserving pretrained knowledge.

Nevertheless, existing methods have two primary drawbacks. First, they use random masks (Hui et al., 2024; Xu et al., 2021) or rely on gradient saliency for selecting task-specific gradient masks (Li et al., 2025). Second, they ignore the explainability of neurons' behaviors from the base model (Choi et al., 2024). Consequently, their redundancy reduction strategies remain less task-sensitive and risk discarding important gradient information. For example, relying solely on high-gradient signals at the batch level can fail to capture neurons that are not gradient-salient but functionally relevant to the downstream task in the pretrained model.

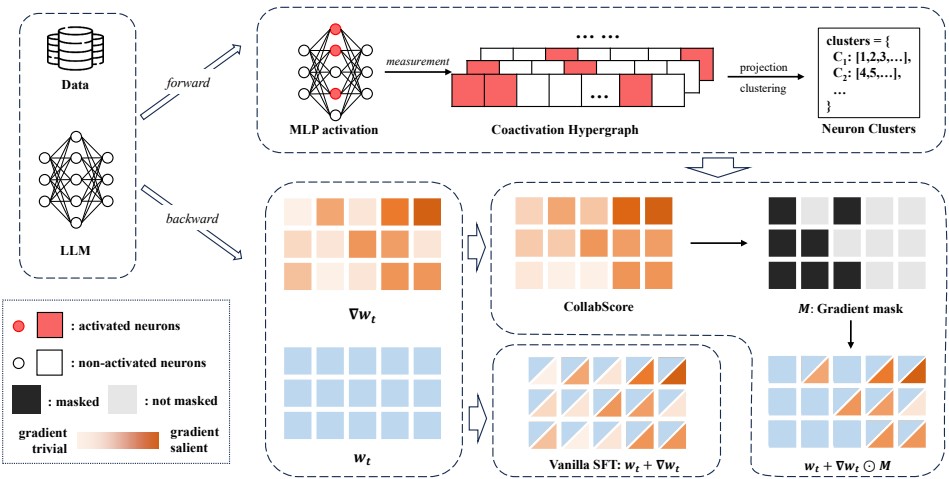

Figure 1: Illustration of our proposed method, `CollabMask`. First, in the forward pass (top-right block), we perform explainable neuron clustering based on co-activation patterns. Training samples are fed into the LLM, and the co-activation of neurons in MLP modules is measured for each layer. These co-activations are used to construct a hypergraph, which is then projected and clustered into explainable neuron groups. Then, in the backward pass during fine-tuning, the parameter gradients $\nabla w_t$ are combined with clustering results to compute the CollabScore, which quantifies the relevance of parameters at the current step $t$. The top-$p\%$ entries of CollabScore are selected via a gradient mask $M$. Applying the mask enables `CollabMask` to selectively update functionally relevant parameters, reducing redundancy compared to the vanilla SFT.

In this paper, we propose `CollabMask` (Neuron Collaboration Gradient Mask Fine-tuning), a novel fine-tuning framework that tackles continual learning challenges from an explainability perspective. `CollabMask` measures neuron co-activation triggered by samples from the training dataset to capture the functional and semantic similarity among neurons, which reflects their collaborative roles in the downstream task. Based on this task-dependent similarity structure, `CollabMask` applies dynamic gradient masks that adapt to each training batch, constraining parameter updates accordingly. By integrating explainability with dynamic masking, `CollabMask` enables targeted fine-tuning ,which more precisely reduces redundant parameter updates.

In detail, `CollabMask` operates in two phases. In the explainable neuron clustering phase, data from the downstream task are used to construct co-activation hypergraphs for MLP layers, which are then projected into collaboration graphs—weighted graphs where vertices represent neurons and edge weights reflect their tendency to be co-activated. Clustering these graphs yields groups of neurons with shared semantic roles, describing the spatial locality of neurons, i.e., subsets that are likely to be useful together for the task. These clusters define task-specific subspaces for fine-tuning. In the masked tuning phase, `CollabMask` dynamically combines collaboration relationships with gradient information to generate batch-specific gradient masks. These masks suppress updates to less relevant neurons while preserving pretrained collaboration patterns, ensuring that functionally important but non-gradient-salient neurons are still updated. Together, these two phases improve fine-tuning by concentrating updates on functionally meaningful neurons.

We conduct extensive experiments on several popular tasks, including medical QA, math, and coding. `CollabMask` is evaluated against multiple baselines—vanilla fine-tuning, LoRA, random masking, highest-gradient masking, and GMT—across three LLMs. On math tasks, `CollabMask` improves accuracy by 4.4% over standard SFT and 2.4% over the previous best method. For coding tasks, it achieves an average improvement of up to 10.5% compared to normal SFT. These results demonstrate that `CollabMask` effectively filters gradient noise during fine-tuning and preserves functionally important neurons, enabling more efficient task adaptation.

Our contribution can be summarized as:

1. We reveal that existing fine-tuning methods fail to effectively leverage the collaborative behavior of neurons, leading to inefficient parameter updates during fine-tuning.

2. We propose `CollabMask`, a method that clusters neurons into functional groups and uses their collaborative behavior to guide gradient masking. This approach reduces unnecessary parameter updates while maintaining neuron collaboration after fine-tuning. We also provide theoretical insights into the effectiveness of this method.

3. We conduct extensive experiments on multiple benchmarks, demonstrating that our method outperforms or competes with representative baselines in various tasks. Additionally, we show that neuron clusters serve as a meaningful explainable unit, opening up potential avenues for future research through cluster evaluations.

## 2 RELATED WORK

**Finetuning strategy** A number of fine-tuning strategies have been proposed to reduce parameter redundancy in LLMs. LoRA (Hu et al., 2021) applies low-rank adaptation to parameters, effectively constraining the hypothesis space to a lower-dimensional subspace. In addition, several gradient masking methods have been developed to restrict gradient updates during training. For example, Child-Tuning (Xu et al., 2021) and HFT (Hui et al., 2024) assign each parameter a fixed probability of being updated in each step, while GMT (Li et al., 2025) accumulates gradients and updates only the top-$p\%$ parameters with the largest absolute gradient values.

**Neuron-level explainability** A major line of research in LLM explainability focuses on understanding the functions and semantics of individual neurons, as well as their influence on model outputs (Sajjad et al., 2022; Zhao et al., 2023). Neuron attribution methods aim to identify critical neurons responsible for specific tasks or knowledge retrieval (Yu & Ananiadou, 2024; Lan et al., 2023; Wang et al., 2022). Another direction is mechanistic interpretability, which studies how groups of neurons across layers form circuits that explain model behavior or specific decisions (Conmy et al., 2023; He et al., 2024). Some works attempt to assign global functional explanations. Examples include linear explanation approaches (Oikarinen & Weng, 2024), neuron graph explanation (Foote et al., 2023), and using LLMs to describe the semantic functions of neurons (Choi et al., 2024). In addition, La Rosa et al. (2023) extends single-neuron analysis beyond high-activation cases, proposing multi-level activation explanations to capture a richer picture of individual neuron behavior.

**Model partitioning** Several researchers have explored partitioning deep neural networks into clusters. For example, Zhang et al. (2022) cluster the MLP layers of large language models (LLMs) and transform the architecture into a Mixture-of-Experts (MoE) structure (Shazeer et al., 2017) to improve inference efficiency. In addition, a line of work has investigated network partitioning to support distributed training, deployment, and scalability (Ranjan et al., 2025; Akintoye et al., 2022; Karadag & Topaloglu, 2025).

## 3 METHODOLOGY

In this section, we provide a comprehensive explanation and the underlying principle of `CollabMask`, as shown in Figure 1. First, we measure the co-activation relation among neurons in MLP layers and construct a co-activation hypergraph. Then, the co-activation hypergraph is projected to the collaboration graph and clustered into neuron groups. Finally, we utilize the clustering result of MLP neurons and the gradient from the backward pass to predict the gradient mask for parameter updates.

### 3.1 OVERVIEW OF COLLABMASK

We begin the introduction of `CollabMask` from the standard stochastic gradient descent (SGD) algorithm and the masked tuning technique, used in Hui et al. (2024); Xu et al. (2021); Li et al. (2025). In SGD, parameters are updated in the negative direction of the gradient, under the assumption that the gradient of the batch-wise local gradient approximates the global loss function. Let $w_t$ denote the parameters at step $t$, $\mathcal{L}(B_t, w_t)$ as the loss function on batch $B_t$, and $\eta$ the learning

rate.The update rule can be described as follows:

$$w_{t+1} = w_t - \eta \frac{\partial \mathcal{L}(batch_t, w_t)}{\partial w_t}. \tag{1}$$

Masked optimization methods such as child-tuning (Xu et al., 2021) and GMT (Li et al., 2025) use a parameter-wise mask $M$ to restrict gradient updates:

$$w_{t+1} = w_t - \eta \frac{\partial \mathcal{L}(batch_t, w_t)}{\partial w_t} \odot M, \tag{2}$$

where the mask $M$ is binary,$M \in \{0, 1\}^{|w_t|}$,with $|w_t|$ denoting the shape of parameters.

The key advantage of `CollabMask` lies in its *explainable* mask selection mechanism that considers neuron collaboration. Comparing with previous methods, `CollabMask` adapts gradient selection by incorporating both batch-level parameter gradients and the collaboration behavior of neuron activations. This allows fine-tuning to focus on functionally relevant parameters, thereby reducing the redundancy in parameter updates more precisely.

### 3.2 EXPLAINABLE NEURON CLUSTERING

To guide fine-tuning with an interpretable neuron collaboration structure, `CollabMask` partitions neurons in each MLP layer into functional clusters based on their co-activation in the base model. This process consists of three steps: (i) constructing a co-activation hypergraph, (ii) projecting the hypergraph into a weighted collaboration graph, and (iii) applying spectral clustering. Additionally, we define a metric evaluating how well the resulting clusters reflect the true co-activation structure.

#### 3.2.1 CO-ACTIVATION HYPERGRAPH CONSTRUCTION

Consider an MLP layer $\ell$ with $N_\ell$ neurons. For an input token $x$ from the downstream dataset $\mathcal{D}$, let $z_{\ell,n}(x)$ denote the activation value after the nonlinear activation function (e.g., SiLU in LLaMA) of neuron $n$. A binary indicator is defined as

$$I_{\ell,n}(x) = \mathbf{1}\big[z_{\ell,n}(x) > 0\big], \quad I_{\ell,n}(x) \in \{0, 1\}. \tag{3}$$

The set of activated neurons for token $x$ forms a hyperedge$E_\ell^{(h)}(x)$:

$$E_\ell^{(h)}(x) = \{n \mid I_{\ell,n}(x) = 1\}. \tag{4}$$

Collecting all such hyperedges yields a co-activation hypergraph$\mathcal{H}_\ell = (V_\ell, \mathcal{E}_\ell)$,where the vertex set $V_\ell = \{1, 2, \ldots, N_\ell\}$ is the set of neurons and the hyperedge set$\mathcal{E}_\ell = \{E_\ell^{(h)}(x) \mid x \in \mathcal{D}\}$ is the set of hyperedges.

#### 3.2.2 PROJECTION TO COLLABORATION GRAPH

Following Li & Milenkovic (2017), we project the hypergraph $\mathcal{H}_\ell$ onto a weighted undirected graph $G_\ell = (V_\ell, E_\ell)$, referred to as the collaboration graph. For a pair of neurons $i$ and $j$, the edge weight is defined as:

$$E_\ell(i, j) = \sum_{i,j \in E_\ell^{(h)}(x) \in \mathcal{E}_\ell} \frac{1}{|E_\ell^{(h)}(x)|^\alpha} R(x), \tag{5}$$

where $E_\ell^{(h)}(x)$ is the hyperedge induced by token $x$, $|E_\ell^{(h)}(x)|$ denotes its cardinality, $\alpha$ is a scaling parameter, and $R(x)$ is a token reweighting factor. Intuitively, $E_\ell(i, j)$ measures the strength of collaboration between neurons $i$ and $j$, adjusted for the size of co-activation and token informativeness.

The parameter $\alpha$ is used to control the relative contribution of hyperedges of different sizes. When $0 < \alpha < 2$, the method emphasizes high-degree hyperedges that represent widespread co-activation patterns, while $\alpha > 2$ gives greater weight to low-degree hyperedges. In our experiments, we set $\alpha = 2.5$ to strike a balance: prioritizing tighter co-activation patterns while also retaining information from broader contexts. More detailed reasoning for this choice is provided in A.1.

**Token-frequency reweighting** A naive projection tends to overemphasize frequent tokens (e.g., special tokens such as eos_token), which often dominate the co-activation hypergraph yet contribute minimally to task-specific semantics. To mitigate this bias, we introduce a rank-based reweighting factor $R(x)$. Specifically, we first compute token frequencies over the training dataset and rank tokens in ascending order of frequency. Each token x is then assigned a rank $rank(x)$, and the reweighting factor is defined as $R(x) = rank(x)^\beta$. In the following experiments, we set $\beta = 1.3$. This formulation ensures that medium-frequency tokens, which are typically more informative for the downstream task, have a proportionally greater influence on the construction of the collaboration graph. Details explained at A.2

### 3.2.3 SPECTRAL CLUSTERING

We then apply spectral clustering to the collaboration graph $\mathcal{G}_\ell$. Let $A_\ell$ be the adjacency matrix of $\mathcal{G}_\ell$ with entries $E_\ell(i,j)$, and let $D_\ell$ be the corresponding diagonal degree matrix. The normalized Laplacian of $\mathcal{G}_\ell$ is

$$\boldsymbol{L}_\ell = \boldsymbol{I} - \boldsymbol{D}_\ell^{-\frac{1}{2}} \boldsymbol{A}_\ell \boldsymbol{D}_\ell^{-\frac{1}{2}}. \tag{6}$$

Following the spectral clustering procedure, we compute the $K$ leading eigenvectors of $L_\ell$ to embed the neurons into a low-dimensional space. The resulting embeddings are then clustered by recursively calling the $k$-means algorithm until the desired effectiveness of clustering is reached. The resulting partition of neurons,

$$\mathcal{C}_\ell = \{C_{1,\ell}, C_{2,\ell}, \ldots, C_{k_\ell,\ell}\}, \quad \bigcup_c C_{c,\ell} = V_\ell, \tag{7}$$

defines clusters of neurons that tend to co-activate. Equivalently, the clustering result can be expressed in the co-cluster relation matrix $\boldsymbol{C} \in \{0,1\}^{N_\ell \times N_\ell}$, where

$$\boldsymbol{C}_\ell(i,j) = \begin{cases} 1, & \text{if } \exists C_{m,\ell} \in \mathcal{C}_\ell, i,j \in C_{m,\ell} \\ 0, & \text{otherwise} \end{cases}, \tag{8}$$

This matrix explicitly encodes whether a pair of neurons belongs to the same cluster. Example results and analysis in A.3.

### 3.3 GRADIENT MASK SELECTION

At each optimization step, we construct a binary mask for each parameter block in MLP layers, where the mask has a pass rate of $p\%$. The mask determines which parameters are updated in the current step. [1] To guide mask selection, we define the CollabScore $Collab\text{-}S(\boldsymbol{w}_t)$ that measures the relevance of parameters $\boldsymbol{w_t}$ at step $t$,

$$Collab\text{-}S(\boldsymbol{w}_t) = (1-\lambda) \cdot |\nabla_{\boldsymbol{w}_t}\mathcal{L}| + \lambda \cdot \boldsymbol{A}(\boldsymbol{w}_t), \tag{9}$$

where $A(\boldsymbol{w}_t)$ is the collaboration activation, and $\lambda$ is the factor that controls the contribution of collaboration activation. It measures the average activation level of neuron clusters and is defined as

$$\boldsymbol{A}(\boldsymbol{w}_t) = |\nabla_{\boldsymbol{w}_t}\mathcal{L}| \cdot Norm(\boldsymbol{C}_\ell), \tag{10}$$

where $Norm$ is the row-wise normalizing operation. Finally, the top $p\%$ of $Collab\text{-}S(\boldsymbol{w}_t)$ scores are assigned a mask value of 1, with the rest set to 0. This procedure yields an explainable, dynamic, and collaboration-aware gradient mask for parameter updates.

## 4 EXPERIMENTS

We conduct extensive experiments to demonstrate that `CollabMask` effectively filters out gradient noise and enhances overall fine-tuning performance.

---

[1] By parameter block, we refer to weight matrices such as up_proj, down_proj, and gate_proj in LLaMA.

### 4.1 EXPERIMENT SETTING

**Datasets.** To evaluate the effectiveness of `CollabMask`, we consider three representative down-stream tasks: mathematics, coding, and medical question answering. For the mathematics task, we fine-tune and evaluate on GSM8K (Cobbe et al., 2021). For the coding task, we use CodeAlpaca (Chaudhary, 2023) for fine-tuning and evaluate with the toolkit Liu et al. (2023) on HumanEval (Chen et al., 2021) and MBPP (Austin et al., 2021). For the medical question answering task, we fine-tune on Medical-o1-Reasoning-SFT (Chen et al., 2024) and evaluate on PubMedQA (Jin et al., 2019).

**Base models.** We select large language models from three different families: DeepSeek (DeepSeek-AI, 2025), Mistral (Jiang et al., 2023), and Qwen (Team, 2025). Specifically, we use DeepSeek-R1-distill-llama-8B, Mistral-v0.1-7B, and Qwen3-8B. All models are base versions without prior task-specific fine-tuning.

**Baselines.** To assess the effectiveness of `CollabMask`, we compare it against the following fine-tuning methods:

1. **SFT**: Standard supervised fine-tuning.

2. **LoRA**: fine-tuning with LoRA (Hu et al., 2021).

3. **Random**: child-tuning (Xu et al., 2021) and half fine-tuning (Hui et al., 2024). Random gradient masking.

4. **Highest**: Updating only the top $p\%$ of gradients in each step.

5. **GMT**: Gradient-Masked Fine-Tuning (Li et al., 2025), which accumulates gradients over batches and updates only the top $p\%$.

For fairness and breadth, we fine-tune all models on all tasks under two mask-ratio $p$ settings. For `CollabMask`, we additionally evaluate different combinations of mask ratio $p$ and cluster activation weight $\lambda$.

### 4.1.1 EXPERIMENT RESULTS

Table 1 shows the performance of three models under various fine-tuning and gradient-mask-selection strategies on medical QA and math tasks, while Figure 2 presents the results of the same fine-tuning techniques applied to two of the models on coding tasks.

**Medical QA.** For the Medical QA task, our proposed `CollabMask` consistently outperforms naive neuron-selection strategies and achieves competitive performance against GMT across three distinct LLMs. At $\lambda = 0.5$, CollabMask achieves 81.5 on Deepseek-llama-8B, 77.5 on Mistral-7B, and 85.5 on Qwen3-8B, yielding an average of 82, which surpasses both SFT (80.0) and LoRA (81.2) as well as random and highest-activation selection (79.3 and 79.2). Increasing $\lambda$ to 0.7 provides marginal gains for some models (e.g., Mistral-7B) but slightly reduces performance on others (e.g., Deepseek-llama-8B), indicating that moderate collaboration-based neuron selection achieves the best balance. Overall, these results highlight that CollabMask effectively leverages co-activation patterns to improve model performance, approaching the performance of GMT.

**Math.** In math tasks, `CollabMask` consistently demonstrates superior performance across different LLMs. As shown in the table, with $\lambda = 0.5$, `CollabMask` achieves the highest scores on Deepseek-llama-8B (47.3) and Qwen3-8B (61.3), while remaining competitive on Mistral-7B (45). Compared to standard SFT and LoRA baselines, which achieve 45/40.7 on Deepseek-llama-8B, 37/35.3 on Mistral-7B, and 58.3/51.3 on Qwen3-8B, `CollabMask` consistently improves results. Against other benchmarks, including Random, Highest, and GMT, `CollabMask` shows notable gains: for example, it surpasses GMT by 3.4% on average across models. When varying $\lambda$ to 0.7, performance remains stable, indicating robustness to hyperparameter changes. Overall, `CollabMask` achieves an average score of 51.2 across the three models, reflecting its ability to effectively leverage collaborative masking to enhance mathematical reasoning in LLMs.

Table 1: Accuracy results (%) of fine-tuned models in medical QA and math tasks. The best results are **bold**, and best in `CollabMask` is underlined. $|\theta|$ denotes the total number of parameters, a measure of the model's size. $p$ is the gradient mask pass rate, and $\lambda$ is the collaboration activation contribution factor, both defined in 3.3

| Medical QA | | | | | | | | CollabMask | |
|---|---|---|---|---|---|---|---|---|---|
| **Model** | $|\theta|$ | **SFT** | **LoRA** | $p$ | **Random** | **Highest** | **GMT** | $\lambda = 0.5$ | $\lambda = 0.7$ |
| Deepseek-llama | 8B | 81.0 | 77.5 | 0.5 | 79.0 | 80.0 | 80.0 | 81.5 | 79.0 |
| | | | | 0.7 | 79.5 | 79.0 | **82.0** | 79.0 | 79.0 |
| Mistral | 7B | 72.5 | 79.5 | 0.5 | 71.0 | 72.5 | 77.0 | 77.5 | 77.5 |
| | | | | 0.7 | 72.0 | 69.0 | **79.0** | 78.0 | 76.0 |
| Qwen3 | 8B | 86.5 | 86.5 | 0.5 | 86.5 | 82.5 | **88.0** | 85.5 | 85.0 |
| | | | | 0.7 | 85.0 | 85.0 | 87.0 | 85.0 | 85.0 |
| Average | – | 80.0 | 81.2 | – | 79.3 | 79.2 | **83.0** | 82.0 | |
| **MATH** | | | | | | | | **CollabMask** | |
| **Model** | $|\theta|$ | **SFT** | **LoRA** | $p$ | **Random** | **Highest** | **GMT** | $\lambda = 0.5$ | $\lambda = 0.7$ |
| Deepseek-llama | 8B | 45.0 | 40.7 | 0.5 | 44.5 | 40.0 | 40.0 | **47.3** | 47.3 |
| | | | | 0.7 | 44.7 | 43.7 | 35.7 | 45.3 | 44.0 |
| Mistral | 7B | 37.0 | 35.3 | 0.5 | 41.3 | 38.3 | **46.3** | 45.0 | 42.7 |
| | | | | 0.7 | 44.7 | 38.3 | 50.7 | 43.0 | 39.3 |
| Qwen3 | 8B | 58.3 | 51.3 | 0.5 | 53.0 | 52.0 | 57.0 | 57.0 | 57.3 |
| | | | | 0.7 | 57.0 | 55.3 | 46.7 | **61.3** | 60.3 |
| Average | – | 46.8 | 42.4 | – | 48.8 | 45.8 | 47.8 | **51.2** | |

**Coding.** In coding tasks, `CollabMask` demonstrates model-dependent improvements. On DeepSeek-R1-distill-llama-8B, all methods—including normal SFT, Random, Highest, GMT, and `CollabMask` with $\lambda = 0.5$ or $0.7$—achieve performance similar to SFT, with minor gains observed on HumanEval (0.6) and HumanEval+ (1.2), while mbpp and mbpp+ show negligible changes. In contrast, on Mistral-7B, `CollabMask` consistently outperforms normal SFT across all benchmarks, with notable gains on mbpp (12.7/13.5) and mbpp+ (7.0/11.4), yielding an average improvement of 10.5%. Although GMT sometimes attains slightly higher scores, `CollabMask` achieves comparable performance while maintaining robustness across different $\lambda$ values, highlighting its effectiveness in enhancing coding ability.

## 5 DISCUSSION

### 5.1 TASK-SPECIFICITY OF COLLABORATION GRAPH.

We evaluate the task specificity of neuron clustering by measuring the similarity between different clusterings of the same LLM. Clustering on the same dataset, even across different subsets, produces highly consistent results across all layers, demonstrating the stability of explainable neuron clustering in capturing semantic features. In contrast, clustering across different datasets yields similar results only in the early layers but diverges in the later ones, suggesting that shallow layers primarily encode general syntactic functions, whereas deeper layers capture task-specific semantic features, consistent with findings from the layer-wise explainability research (Chuang et al., 2024). Evaluation metric and detailed results stated in A.4.2.

### 5.2 LIMITATIONS

The inconsistent performance of `CollabMask` compared to prior methods may stem from several factors. First, similar to previous methods, `CollabMask` still heavily uses gradient saliency in mask selection, even though with neuron collaboration relation to enhance it. Second, the co-activation behavior of neurons is measured at the token level, while the loss function is optimized at the batch level. This mismatch may reduce the effectiveness of the collaboration graph in guiding the

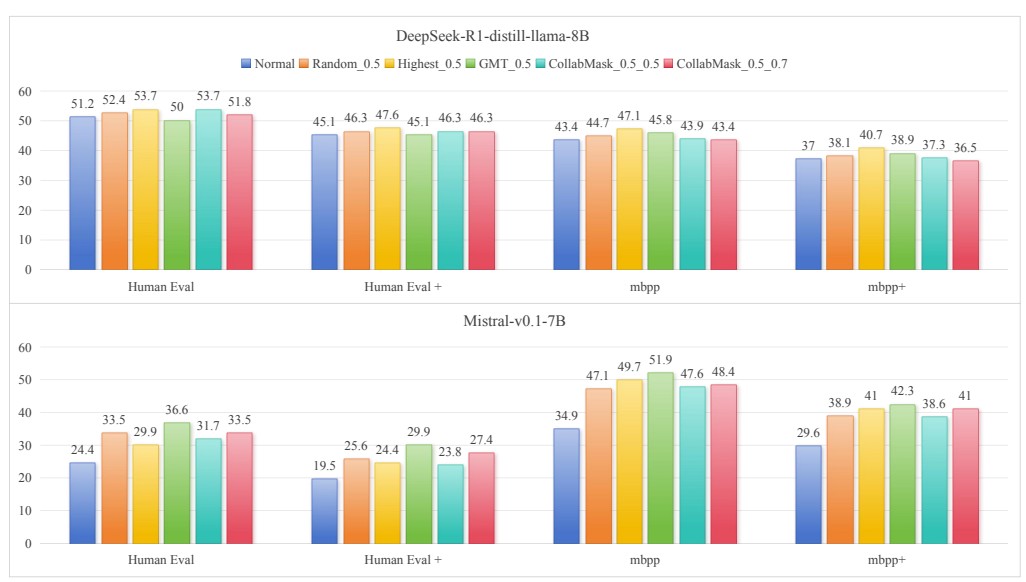

Figure 2: Pass@1 result for coding tasks.

gradients of parameters. Third, the recursive clustering process applied to the collaboration graph can introduce substantial noise into the final mask selection. Employing soft clustering methods or directly applying the collaboration graph to gradients without clustering may mitigate this issue and improve fine-tuning performance.

`CollabMask` introduces computational overhead due to co-activation hypergraph construction, graph projection, and spectral clustering steps. In addition, our current method focuses on MLP layers; neuron collaboration relations within attention modules remain underexplored. Another limitation is that we treat layers as independent, whereas cross-layer neuron collaborations and causal relations likely exist, as suggested by mechanistic interpretability research (Conmy et al., 2023).

### 5.3 FUTURE DIRECTIONS

Future work may explore several directions. First, `CollabMask` could be extended to preference optimization frameworks, such as Direct Preference Optimization (DPO) (Rafailov et al., 2023). Second, the framework may be adapted for mixture-of-experts (MoE) models and extended to attention modules. Finally, exploring continuous representations of neuron collaboration could provide a more flexible mechanism for integrating neuron collaboration relations into fine-tuning.

## 6 CONCLUSION

In this paper, we introduced `CollabMask`, a fine-tuning method that leverages neuron collaboration—groups of neurons co-activating for task-specific functions—to build collaboration-aware gradient masks. By identifying functionally important parameter groups from pretrained models, `CollabMask` preserves collaboration patterns, reduces redundant updates, and mitigates overfitting, leading to improved performance and generalization. Experiments show that `CollabMask` outperforms normal SFT by up to 10.5%. Future work will focus on obtaining more stable and interpretable neuron groups and extending neuron collaboration analysis to attention mechanisms and mechanistic interpretability.

# 7 REPRODICIBILITY STATEMENT

All experiments in this paper are implemented in PyTorch 2.8 with CUDA 12.4 on NVIDIA A100 GPUs. The codebase, including training and evaluation scripts, will be made publicly available, and an anonymous link will be provided during the review process.

Models are trained using the Adam optimizer with a learning rate of 1e-5, batch size 32, weight decay 0.01, and a maximum of 3 epochs, with validation performed at each epoch and the best model retained as the final model. For all datasets, reordering is disabled to ensure reproducibility. For the MathQA and math tasks, we use 3,000 samples for training and 1,000 for validation. For the coding task, 4,000 samples are used for training and 1,000 for validation.

The DeepSeek-R1-distilled-llama-8B and Mistral-7B models are fine-tuned using 4 A100 GPUs, while the Qwen3-8B model is fine-tuned on 6 A100 GPUs. Each training epoch takes approximately 20 minutes. For the explainable neuron cluster experiments, hypergraph projection, eigenvector approximation, and clustering take around 5 minutes per layer.

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

## A  APPENDIX

### A.1  THEORETICAL FOUNDATION OF PROJECTION TO COLLABORATION GRAPH

Following the Inhomogeneous Hypergraph Clustering (InH) algorithm by Li & Milenkovic (2017) and its related notation, we demonstrate the consistency of our projection method in `CollabMask` pipeline and highlight its advantage in handling hyperedges with small degrees.

#### A.1.1  PROJECTION OF HYPERGRAPHS IN INH

In InH, it is proved that using the prescribed method, a hypergraph can be projected onto a latent graph, and clustering on this latent graph can approximate the clustering on the original hypergraph.

Formally, let $\mathcal{H} = (V, E)$ denote a hypergraph. For each hyperedge $(e, w_e)$, the cost of cutting $e$ into two disjoint subsets $S \subset e$ and $e \setminus S$ is given by $w_e(S)$. A weight function $w_e(\cdot)$ is called consistent if it satisfies $w_e(S) = w_e(e \setminus S)$.

For each InH-hyperedge $(e, w_e)$, the InH algorithm requires a projected complete subgraph $G_e = (V^{(e)}, E^{(e)}, w^{(e)})$ to represent $(e, w_e)$, where

$$V^{(e)} = e, \quad E^{(e)} = \{\{v, \tilde{v}\} \mid v, \tilde{v} \in e, v \neq \tilde{v}\}. \tag{11}$$

The goal is to find edge weights $w_{v\tilde{v}}^{(e)}$ such that

$$w_e(S) \leq \sum_{v \in S, \tilde{v} \in e \setminus S} w_{v\tilde{v}}^{(e)} \leq \beta^{(e)} w_e(S), \tag{12}$$

where $\beta^{(e)}$ is a constant.

After solving for $w^{(e)}$, InH constructs a complete weighted graph $\mathcal{G} = (V, E_o, w)$, where $V$ is the set of vertices of the hypergraph, $E_o$ is the complete set of edges, and the edge weights are computed as

$$w_{v\tilde{v}} = \sum_{e \in E} w_{v\tilde{v}}^{(e)}. \tag{13}$$

### A.1.2 IMPLEMENTATION OF PROJECTION IN COLLABMASK

In `CollabMask`, when defining the cost of cutting a co-activation hypergraph, it is natural to use a non-constant cost function so that cutting out a small fraction of a hyperedge incurs a lower penalty, while keeping the majority of vertices in the same cluster. We define the cost function of a co-activation hyperedge, which is obviously consistent by definition, as follows,

$$w_e(S) = \frac{|S| \cdot (|e| - |S|)}{|e|^\alpha}, \tag{14}$$

where $\alpha$ is a constant parameter that controls how the hyperedge degree affects clustering, thereby prioritizing low-degree hyperedges in subsequent steps of the `CollabMask` pipeline. When $|S| = \frac{|e|}{2}$, the maximum cost of a hyperedge is reached:

$$\max\ w_e(S) = \frac{1}{4|e|^{\alpha-2}}. \tag{15}$$

Under the current setting $\alpha = 2.5$, this becomes

$$\max\ w_e(S) = \frac{1}{4\sqrt{|e|}}, \tag{16}$$

which emphasizes hyperedges of smaller degree.

The edge weights in the projected graph are set as $w_{v\tilde{v}}^{(e)} = \frac{1}{|e|^\alpha}$, which satisfy the approximation requirement in equation 12:

$$\sum_{v \in S, \tilde{v} \in e \setminus S} w_{v\tilde{v}}^{(e)} = \frac{|S| \cdot (|e| - |S|)}{|e|^\alpha} = w_e(S). \tag{17}$$

Finally, the projection graph is constructed following equation 13, yielding the collaboration graph (similar to equation 5, but without token reweighting):

$$E_\ell(i, j) = \sum_{i,j \in E_\ell^{(h)}(x) \in \mathcal{E}_\ell} \frac{1}{|E_\ell^{(h)}(x)|^\alpha}. \tag{18}$$

### A.2 TOKEN-FREQUENCY REWEIGHTING

In linguistics, the frequency of words is often analyzed as a function of their frequency rank, following Zipf's law (Zipf, 1949). Empirically, words serving primarily syntactic purposes (e.g., "the") typically appear among the highest-frequency ranks. In LLMs, a similar imbalance occurs: the `padding_token` often dominates the input, sometimes accounting for nearly half of the tokens, due to the usual 's padding-to-max-length strategy.

As a result, most hyperedges $E_\ell^{(h)}(x)$ in the co-activation hypergraph are induced by high-frequency tokens that contribute relatively little semantic information. To mitigate this bias, we introduce a token-frequency-based reweighting factor

$$R(x) = \text{rank}(x)^\beta, \tag{19}$$

where $\beta = 1.3$ in the current implementation, and additionally remove hyperedges induced by extremely high-frequency tokens (in our current setting, the top 3 tokens).

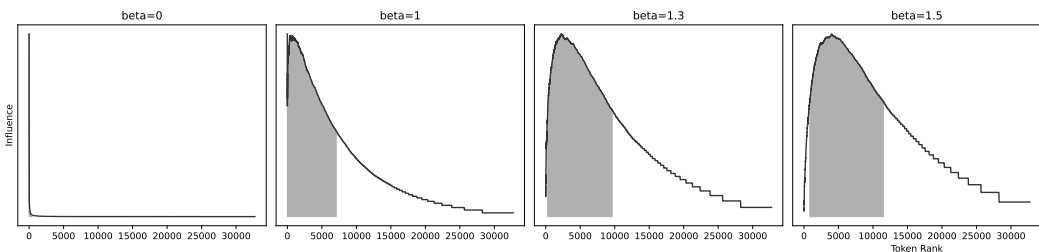

Figure 3: The influence of tokens with different frequency under different reweighting factor $\beta$. The dark area is the minimum token rank range that covers more than 60% of the total influence under the $\beta$ setting. Measured in Medical-o1-reasoning-SFT-en (Chen et al., 2024).

Here, the influence of a token is measured as the normalized product of its frequency and its reweighting factor. This adjustment reduces the dominance of high-frequency, low-semantic tokens while amplifying the relative contribution of medium- and low-frequency tokens. The resulting influence distribution across tokens of different frequencies is illustrated in Figure 3.

### A.3 ADDITIONAL RESULTS OF SPECTRAL CLUSTERING

During spectral clustering, we use a $k$-nearest-neighbor approximation of the adjacency matrix of the projected collaboration graph. This sparsification accelerates the computation of the eigenvectors of the corresponding normalized Laplacian. We record the eigenvector embeddings of the Laplacian and subsequently reduce their dimensionality to 2D using the t-SNE method for visualization. The resulting embeddings and clustering results are shown in Figure 4.

As observed, the neuron embeddings measured via co-activation do not exhibit clear or strictly separable clusters. Furthermore, enforcing clustering to carry at least 4 bits of information by recursively partitioning the largest clusters can introduce abrupt changes in cluster assignments for relatively small variations in the embeddings. This sensitivity may lead to noisy neuron groupings and, consequently, degrade the final fine-tuning performance.

### A.4 EVALUATION OF CLUSTERING

#### A.4.1 EVALUATION OF CORRESPONDENCE TO CO-ACTIVATION

To evaluate whether a clustering $\mathcal{C}_\ell$ faithfully reflects the neuron collaboration structure, we define the following conditional probability metric:

$$\text{Score}(\mathcal{C}_\ell, x) = \Pr[\mathcal{C}_\ell(i) = \mathcal{C}_\ell(j) \mid I_{\ell,i}(x) = 1, I_{\ell,j}(x) = 1], \tag{20}$$

where $x$ is a token sampled from the unseen section from the same dataset. This probability can be equivalently expressed in closed form as

$$\text{Score}(\mathcal{C}_\ell, x) = \frac{\sum_{c=1}^{k_\ell} |HE_\ell(x) \cap C_{c,\ell}|^2}{|HE_\ell(x)|^2}. \tag{21}$$

This measures the probability that two simultaneously activated neurons fall into the same cluster.

As a baseline, the expected probability under random assignment (the null model) can be computed as

$$\text{Null}(\mathcal{C}_\ell) = \sum_{c=1}^{k_\ell} \frac{|C_{c,\ell}|^2}{N_\ell^2}. \tag{22}$$

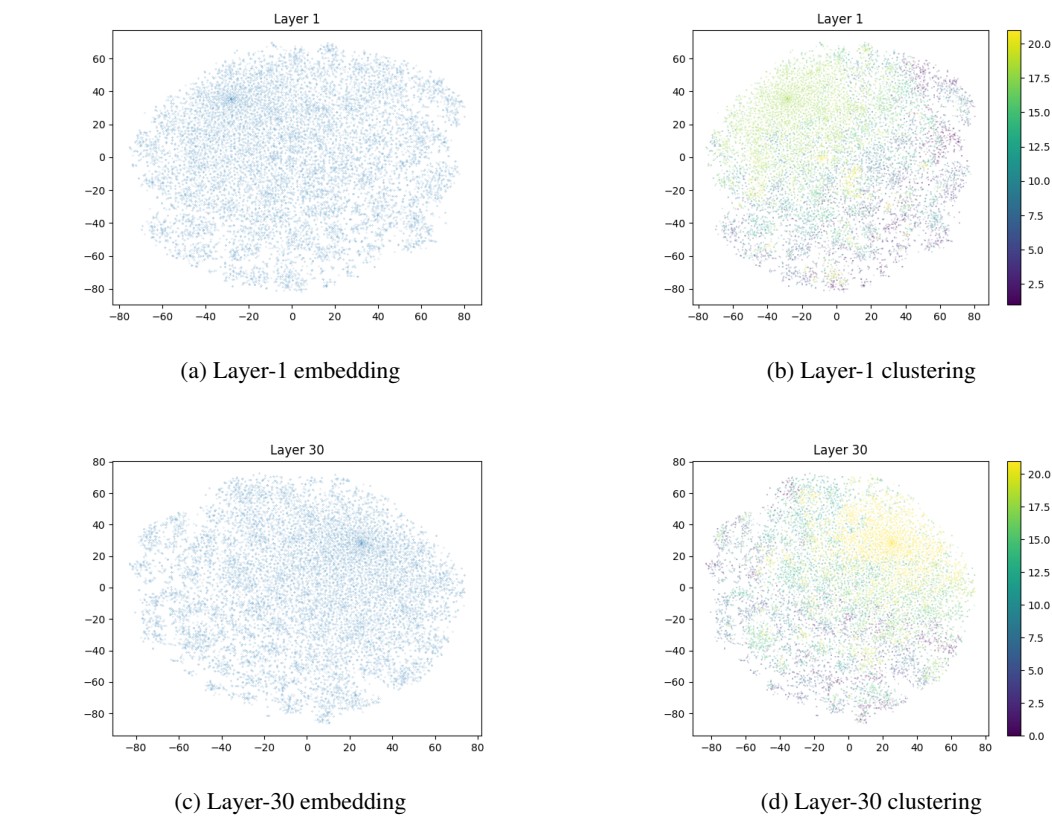

(a) Layer-1 embedding        (b) Layer-1 clustering

(c) Layer-30 embedding        (d) Layer-30 clustering

Figure 4: Example results from spectral clustering of deepseek-R1-distilled-llama-8B on dataset GSM8k

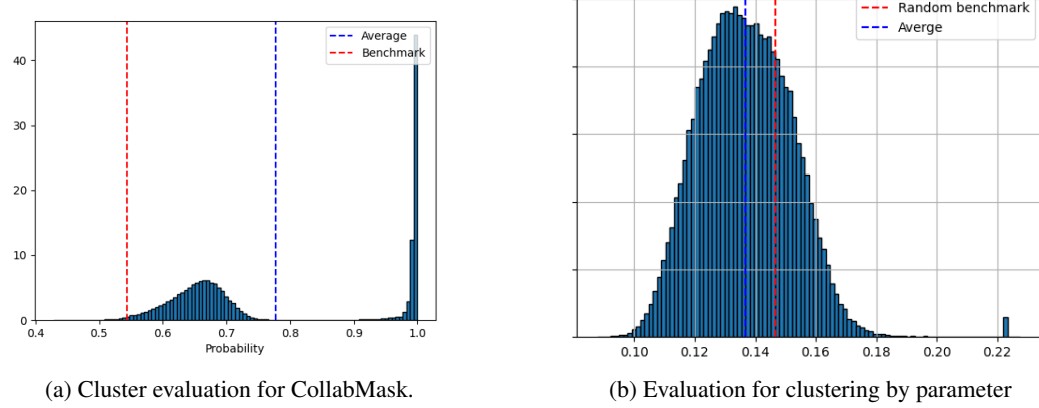

(a) Cluster evaluation for CollabMask.    (b) Evaluation for clustering by parameter

Figure 5: Cluster evaluation results

In Figure 5, we present an example evaluation of how clustering corresponds to co-activation. The red dashed line represents the Null benchmark defined in Equation 22, and the blue dashed line represents the average probability over input tokens. In Figure 5a, we show the clustering evaluation result of layer 14 of `deepseek-R1-distilled-llama-8B`, without enforcing recursive clustering. In contrast, a baseline method Figure 5b presents the evaluation result for the correspondence between co-activation and clustering based on parameters. In this setting, neurons are clustered using the $k$-means algorithm with cosine similarity as the distance metric, applied to their associated parameter vectors in the `up_proj` parameter block.

Note that the two graphs have different $x$-axis scales. Overall, the results demonstrate that explainable neuron clustering more accurately reflects neuron collaboration behaviors in the given downstream dataset.

### A.4.2 EVALUATION OF TASK SPECIFICITY

We also evaluate how the neuron clustering results reflect task specificity. To quantify this, we define a metric that measures the similarity between two partitions of the same neuron set. Specifically, we randomly sample two distinct neurons and compute the probability that the two clustering results agree on whether the pair is co-clustered or not co-clustered. Formally, let $\mathcal{C}_1$ and $\mathcal{C}_2$ be two partitions of the neuron set $V$.

Using this metric, we compare the difference between Case 1 and the following scenarios: 1. `deepseek-llama-8b` clustered on `medical-o1-reasoning-sft`, 2. the same model clustered on `gsm8k`, and 3. the same model clustered again on `medical-o1-reasoning-sft`, but using a different subset of samples.

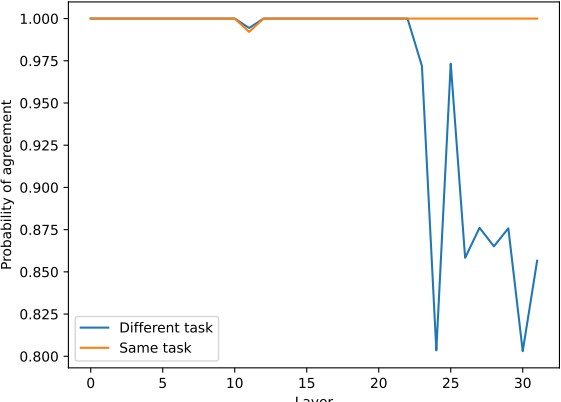

Figure 6: Similarity between partitions .

As shown in Figure 6, the similarity between case 1 and case 3 remains close to 1 across all layers, indicating that the clustering results effectively and stably capture the dataset's semantic features. The similarity between case 1 and case 2 is high in the early layers but diverges substantially in the later layers, suggesting that neuron clustering becomes increasingly task-specific in deeper layers. This finding supports the view that shallow layers in LLMs primarily encode syntactic functions (Chuang et al., 2024), resulting in minimal differences across datasets, while deeper layers capture semantic functions, yielding distinct clustering outcomes across datasets but consistent results within subsets of the same dataset.

### A.5 USE OF LLMS

We used large language models (LLMs) to assist with the preparation of this manuscript. Specifically, LLMs were employed to help with text editing, grammar refinement, and improving readability of certain sections. All technical ideas, experimental design, implementation, and analysis were conducted by the authors.

