# OpenReview forum: "CollabMask: Explainable Neuron Collaboration with Gradient Masks for LLM Fine-Tuning"
_ICLR.cc/2026/Conference — ICLR 2026 Conference Withdrawn Submission_

### Official Review · Reviewer_pKjw · 2025-10-23

**Soundness:** 2
**Presentation:** 2
**Contribution:** 2
**Rating:** 4
**Confidence:** 4

**Summary:**

The maunscript under review tackles redundant updates in LLM fine-tuning by exploiting *neuron collaboration*—groups of co-activated neurons—to build explainable, task-aware gradient masks that focus updates on functionally relevant parameters. The method constructs a co-activation hypergraph from MLP activations on downstream data, projects it to a collaboration graph, clusters neurons into functional groups, and uses these groups to guide masking. During training, it combines batch gradients with cluster activations into a dynamic CollabScore and selects the top p% entries, preserving pretrained collaboration patterns while suppressing noisy updates. Furnished experimental results include that across math (GSM8K), coding (HumanEval/MBPP), and medical QA (PubMedQA) on DeepSeek, Mistral, and Qwen models, CollabMask improves over standard SFT and competitive baselines while yielding interpretable neuron groups.

**Strengths:**

1. The paper introduces a fresh angle on masked fine-tuning by centering mask selection on neuron collaboration: it builds co-activation hypergraphs from downstream data, projects them to collaboration graphs, clusters neurons, and then fuses cluster activations with batch gradients to form a dynamic, explainable mask.

2. Empirically, three model families (DeepSeek, Mistral, Qwen), three task types (math, coding, medical QA), and strong baselines (SFT, LoRA, random, highest, GMT) with consistent training settings across mask ratios and λ values are tested. Results tables and figures substantiate gains and robustness. Reproducibility details (optimizer, batch sizes, epochs, hardware) are spelled out.

3. The method is communicated cleanly via a step-by-step overview and the paper’s structure make it easy to follow.

4. Treating neuron clusters as explainable functional units elevates interpretability from post-hoc analysis to an active driver of training.

**Weaknesses:**

1. Not SOTA and often not best-in-class against close baselines: On Medical QA, CollabMask trails GMT on average (82.0 vs 83.0), and on coding the text concedes “GMT sometimes attains slightly higher scores,” with DeepSeek showing essentially SFT-level results across methods; the clear win is mostly on math (Avg 51.2 vs GMT 47.8). That mix of outcomes doesn’t establish SOTA masked tuning across tasks or models.

2. Baseline coverage and tuning fairness are too thin for a SOTA claim: The baseline set is limited to SFT, LoRA, Random, Highest, and GMT; modern PEFT variants widely used in practice (e.g., quantization-aware adapters, low-rank variants beyond vanilla LoRA) are absent. Moreover, “for fairness” all models use two mask-rate p settings, but CollabMask gets extra tuning over both p and λ—giving it a hyperparameter search advantage over GMT and others. Add stronger PEFT baselines and match hyperparameter search budgets across methods.

3. The core mechanism is heavily heuristic and empirically fragile: Several critical dials are fixed without sensitivity analysis: α=2.5 in the projection, β=1.3 plus removal of the top-3 frequent tokens for reweighting, binary activation indicators 1[z>0], recursive clustering to a “4 bits of information” target, and kNN sparsification. The appendix explicitly notes embeddings “do not exhibit clear or strictly separable clusters,” and the recursion can cause abrupt cluster flips—noise that plausibly explains inconsistent wins. Provide ablations over α, β, k, recursion criteria, and binarization vs magnitude/quantile activations; also report per-layer cluster counts/sizes.

4. Statistical rigor is insufficient to claim robust gains: Tables/figures present single numbers without confidence intervals or multi-seed variation, and the Reproducibility Statement uses small train sets (e.g., 3k math samples; 4k coding) that amplify variance. Run 3–5 seeds with mean±95% CI and add additional datasets per domain (e.g., beyond GSM8K and PubMedQA) to substantiate generalization.

5. “Explainability” remains correlational, not causal: The evaluation measures whether co-activated pairs co-cluster and shows within-dataset stability; however, that’s correlation. There are no causal interventions (targeted cluster ablations/freezes or activation patching) tying clusters to behavior.

**Questions:**

1. Did GMT, Highest, and Random receive the same hyperparameter search budget as CollabMask over the mask rate p and the λ tradeoff?
2. When you fix α=2.5 in the hyperedge projection, how sensitive are results to this choice?
3. Comments from the weaknesses section.

---

### Official Review · Reviewer_kyvA · 2025-10-31

**Soundness:** 2
**Presentation:** 2
**Contribution:** 2
**Rating:** 2
**Confidence:** 4

**Summary:**

CollabMask proposes an explainable neuron collaboration mechanism for gradient masking during large language model (LLM) fine-tuning.
The method aims to reduce redundant parameter updates by constructing co-activation hypergraphs from neuron activations, clustering neurons into functional groups, and using these clusters to guide gradient masking through a collaboration-aware scoring function.
This direction is conceptually sound: it targets a fundamental weakness of prior gradient-masking strategies, which assume neuron independence and ignore functional co-activation structures.

**Strengths:**

The paper correctly identifies the lack of neuron interaction modeling in existing fine-tuning techniques.
By incorporating a co-activation prior, CollabMask introduces interpretability into the fine-tuning process: parameter updates are linked to identifiable neuron clusters derived from the collaboration graph.
Furthermore, the use of a Laplacian-based spectral clustering framework provides an implicit geometric interpretation of functional collaboration, which can be a proper foundation for future collaboration geometry-aware optimization.
The conceptual motivation, to improve fine-tuning efficiency by preserving cooperative neural subspaces, is original and valuable.

**Weaknesses:**

Despite its promising motivation, several aspects of the proposed method limit its effectiveness and, in some cases, contradict its stated goals:

1. **Correlation restricted to positive activations.**
    The co-activation measure relies on a binary indicator $I_{\ell,n}(x) = \mathbf{1}[z_{\ell,n}(x) > 0]$,
    meaning only positive activations contribute to the collaboration graph.
    This assumption neglects neurons with negative activations, which often encode inhibitory or complementary signals.
    As a result, the constructed hypergraph reflects only the frequency of simultaneous positive activations rather than the full correlation structure of neuronal behavior.

2. **Lack of subspace-level collaboration understanding.**
    Treating collaboration as coincidence of positive activation fails to capture directional or subspace geometry among neuron vectors.
    Two neurons with positive activations but opposite representational orientations can behave orthogonally or even destructively when combined, invalidating the assumption that co-activation implies cooperation.
    True collaboration should depend on alignment in its weight representational space, not simply activation coincidence.

3. **Static collaboration structure.**
    The collaboration pattern $C_\ell$ is computed once prior to fine-tuning and kept fixed throughout training.
    However, both model parameters and input distributions change during fine-tuning, meaning the actual co-activation topology may drift significantly.
    Consequently, a static mask may not only fail to reflect true neuron collaboration but may also distort gradient flow by constraining updates in directions inconsistent with the model's evolving representational geometry.
    This contradicts the core motivation of improving task adaptation while preserving meaningful collaboration.

4. **Marginal empirical improvements.**
    Reported results show only modest performance gains with even worse performance to its baselines (GMT) in some task (Medical QA., etc)
    Given the method's complexity, these small improvements suggest that the current algorithm does not effectively capture functional neuron relationships. This empirical evidence thus reflects a misalignment between the theoretical motivation and the actual implementation.

**Questions:**

1. How meaningful is ``collaboration'' if it ignores both sign and directional correlations among neuron activations?
2. Why is the collaboration graph static? Could an online or periodically updated co-activation structure better capture evolving neuronal dependencies?
3. Can collaboration be more faithfully modeled as a subspace geometry problem, where neurons share low-dimensional manifolds or Laplacian eigenstructures rather than discrete co-activation counts?
4. Given the marginal gains, do the derived neuron clusters genuinely reflect semantic or functional groups, or are they artifacts of binarized activations?

---

### Official Review · Reviewer_QtBs · 2025-11-02

**Soundness:** 1
**Presentation:** 2
**Contribution:** 2
**Rating:** 4
**Confidence:** 3

**Summary:**

This paper introduces CollabMask, a neuron-level fine-tuning method for large language models (LLMs). The core idea is to leverage neuron co-activation clustering to guide parameter updates via group-based gradient masking. The motivation stems from the observation that neurons tend to be co-activated as functional groups during forward propagation for specific tasks. CollabMask operates in two stages: (1) Pre-tuning analysis, where the model’s forward activations on downstream task data are used to measure co-activation patterns across MLP neurons, construct a weighted graph, and perform clustering; (2) Fine-tuning, where a dynamic mask operation is applied. Combining gradient saliency, CollabScore derived from the clustered groups is used to pick&update only the most collaborative neurons. Experiments on math reasoning, coding, and medical QA tasks demonstrate that CollabMask outperforms SFT, LoRA, and the gradient masking baseline GMT.

**Strengths:**

1. Novel concept of neuron co-activation. Modeling neuron interactions as a hypergraph and clustering them to guide fine-tuning is original and valuable. It goes beyond existing masking-based PEFT methods, which typically ignore structural dependencies between parameters, providing a fresh perspective for efficient parameter tuning.
2. Comprehensive evaluation setup. The authors test across diverse tasks (math, code, medical QA) and model families (DeepSeek, Mistral, Qwen), with strong and representative baselines including full fine-tuning, LoRA, and GMT. This breadth enhances the credibility of the results.
3. The paper is well-organized and easy to follow, with a logical narrative from motivation to implementation and evaluation.

**Weaknesses:**

1. Limited and unstable performance gains. Although the paper reports a 2.4% improvement on math tasks, CollabMask sometimes underperforms GMT on medical QA and code generation. The improvements are not consistently dominant.
2. Heuristic pipeline with unjustified hyperparameter settings. The method introduces several heuristic design choices. For example, parameter $\alpha$ controls the contribution of hyperedges of different sizes. While Appendix A.1 offers theoretical reasoning, empirical validation (e.g., varying $\alpha$) would strengthen the justification, similar to the intuitive analysis shown for token reweighting in Appendix A.2.
3. Unclear computational overhead. The paper mentions clustering takes ~5 minutes per layer (Section 7), but lacks context on model size and dataset scale. How does this scale to larger models with more layers? A quantitative comparison of efficiency vs. GMT would clarify practical trade-offs.
4. Limited interpretability analysis. Although the title emphasizes interpretability, the paper provides only heuristic discussion. Beyond quantitative clustering metrics, qualitative examples would be valuable. For instance, showing whether specific neuron clusters correspond to identifiable functions (e.g., numerical processing or arithmetic).

**Questions:**

CollabMask performs best on mathematical reasoning but not on other tasks. Do the authors hypothesize that neuron collaboration patterns differ fundamentally across domains? What characteristics of math tasks make them particularly suited to this approach?

---

### Official Review · Reviewer_Ck3p · 2025-11-04

**Soundness:** 1
**Presentation:** 2
**Contribution:** 2
**Rating:** 2
**Confidence:** 4

**Summary:**

The paper proposes CollabMask, a method that leverages sparse masked updates at the neuron level to perform task-specific adaptation on large language models. Specifically, the paper claims to make progress with respect to current literature by considering neuron collaboration (defined as co-activation of neurons for a specific task) when choosing which parts of the model to fine-tune. Based on co-activations of the neurons in each MLP layer, CollabMask groups neurons by constructing a collaboration graph and applying spectral clustering. Experiments are run on Medical QA, Math and Coding benchmarks with open source LLMs (Deepseek, Mistral, Qwen), comparing against several fine-tuning paradigms.

**Strengths:**

- The collaboration-aware masking idea that goes beyond using gradients is interesting and checking collaboration at neuron level is intuitive and makes sense, given recent results in interpretability of LLMs (eg. seeing works from Nanda et al., Bau et al. that show LLM pre-training results in specialized neurons for specific tasks).

- Some gains are present in the experimental results, suggesting that the approach can help in specific cases (for instance, it seems when collaboration structure aligns with the task).

**Weaknesses:**

Although the premises and the general framing and methodology of the paper are interesting, there are several shortcomings that I feel are too extensive at this time to be addressed. Hence, my recommendation for rejection. In detail,

**W1.**

The text frames the method as reducing redundant updates, which presumably situates the contribution into the efficiency literature. Yet, the paper does not reports any compute/memory/wall clock benchmarks versus some baselines. Moreover, it also introduces extra per-layer preprocessing which apart from the note in the reproducibility statement it is not benchmarked in terms of compute/memory usage.

---------------

**W2.**

It is unclear what exactly is explainable in CollabMask. If the authors refer to the mask calibration strategy, I'd say that it is a rather opaque process, given that clustering not necessarily constitutes an explanation. Otherwise, if the authors refer with the term "explainable" to the fact that neurons co-activate then I'm unsure how co-activation inherently ties to explainability.

In both cases, the problem is that correlated signals are not inherently causal [1], underscoring a potential conceptual and fundamental limit of the chosen metric for assessing collaboration between neurons.

---------------

**W3.**

The experimental results highlight some clear limitations of the proposed approach and call into question the rigor of the evaluation:

- Several results show that even Random masks match or exceed CollabMask, with limited discussion of failure modes. For instance, is CollabMask underperforming in some cases due to the pre-training scheme? Are there maybe other factors influencing the performance of the proposed approach?

- The number of runs is not specified. Given the tight clustering of results it would be advisable to report mean and standard deviations of multiple runs.

- The paper lacks an ablation study on how much (and which) data is needed to perform proper gradient mask calibration.

- It is unclear whether adaptation methods converge during fine-tuning: for instance, on Medical QA it is counterintuitive that LoRA performs better than standard full fine-tuning, given it updates far less parameters.

---------------

**W4.**

There is some partial methodological overlap with previous literature [2] that should be acknowledged and contrasted.

---------------

**Minor:**
- In the text, there are some typos in terms of missing spaces between words (eg. check L162 after period, L172 after commas, L198 before the $\mathcal{E}$ etc.)

---------------

**_References:_**

[1] Zhao, Haiyan, et al. "Explainability for large language models: A survey." ACM Transactions on Intelligent Systems and Technology 15.2 (2024): 1-38.

[2] Tan, Shaomu, Di Wu, and Christof Monz. "Neuron Specialization: Leveraging intrinsic task modularity for multilingual machine translation." EMNLP, 2024.

**Questions:**

Thanking in advance for their response, I'd kindly invite the authors to address the points raised in the Weaknesses section of this review.

---

### Note · Authors · 2025-11-12

I have read and agree with the venue's withdrawal policy on behalf of myself and my co-authors.